# Prevalence of COVID-19 Vaccination among Medical Students: A Systematic Review and Meta-Analysis

**DOI:** 10.3390/ijerph19074072

**Published:** 2022-03-29

**Authors:** Romana Ulbrichtova, Viera Svihrova, Jan Svihra

**Affiliations:** 1Department of Public Health, Jessenius Faculty of Medicine in Martin, Comenius University in Bratislava, Mala Hora 11149/4B, 03601 Martin, Slovakia; viera.svihrova@uniba.sk; 2Clinic of Urology, Jessenius Faculty of Medicine in Martin, Comenius University in Bratislava, Kollarova 2, 03659 Martin, Slovakia; jan.svihra@uniba.sk

**Keywords:** COVID-19, prevalence, vaccination, medical students, meta-analysis

## Abstract

The aim of this meta-analysis was to evaluate the prevalence of COVID-19 vaccination among medical students worldwide. Three electronic databases, i.e., PubMed, Scopus, and Web of Science (WoS), were used to collect the related studies according to the Preferred Reporting Items for Systematic Reviews and Meta-Analyses (PRISMA) guidelines. The study population included undergraduate medical students who had already been vaccinated reported in original articles published between January 2020 and December 2021. The heterogeneity of results among studies was quantified using the inconsistency index I^2^. Publication bias was assessed by using Egger’s test. Six cross-sectional studies with 4118 respondents were included in this study. The prevalence of COVID-19 vaccination was 61.9% (95% CI, 39.7–80.1%). There were no statistical differences between gender and vaccination acceptance, 1.038 (95% CI 0.874–1.223), and year of study and vaccination acceptance, 2.414 (95% CI, 0.754–7.729). The attitudes towards compulsory vaccination among healthcare workers can be determined by a prevalence of 71.4% (95% CI, 67.0–75.4%). The prevalence of COVID-19 vaccination among medical students was at a moderate level. Placing a greater emphasis on prevention seems essential in the medical curriculum.

## 1. Introduction

The world is currently facing a fourth wave of COVID-19, as, on 26 November 2021, the World Health Organisation (WHO) designated the variant B.1.1.529 a variant of concern, named Omicron [1]. Currently, most countries are administering booster doses of COVID vaccines [2]. The implementation of global and national vaccination programmes in combination with nonpharmaceutical interventions are the only available tools to end the COVID-19 pandemic [3].

The COVID-19 pandemic requires a great deal of discussion about vaccination, especially among healthcare workers (HCWs). Last but not least, medical students also play a key role [4,5,6]. To prevent the healthcare system from being overwhelmed, countries have decided to involve medical students as volunteers or to carry out their practical training in hospitals [7]. HCWs and medical students are at a greater risk for COVID-19 exposure. Educating medical students on vaccination issues is another aspect that is also very important for their subsequent employment. Their attitudes and opinions can significantly influence family and friends, as they are considered competent persons and important sources of information for the general public [8].

In several countries, medical students were vaccinated in the early phases of vaccination [9,10]. Slovakia prioritised in the first phase the vaccination of HCWs and medical students who were in contact with COVID-19 patients [11].

Vaccine hesitancy varies depending on the place, time, and perceived nature of the behaviour community [12]. In the current ongoing pandemic, it is essential to monitor and analyse the prevalence and attitudes towards vaccination against COVID-19 among medical students.

The aim of this meta-analysis was to evaluate the prevalence of COVID-19 vaccination among medical students worldwide.

## 2. Materials and Methods

### 2.1. Protocol and Registration

The study was performed according to the Preferred Reporting Items for Systematic Reviews and Meta-Analyses (PRISMA) guidelines (Appendix A) [13]. The study protocol was registered with the international prospective register of systematic reviews (PROSPERO) (CRD42022301854).

### 2.2. Eligibility Criteria

The inclusion criteria were: (1) observational studies (including cross-sectional studies) published in peer-reviewed journals; (2) undergraduate medical students in any year of study who have been enabled to be vaccinated against COVID-19; (3) already vaccinated medical students; (4) provided raw data. The exclusion criteria were as follows: (1) nonrelevant articles, reports, editorials, letters, and protocols; (2) duplicates; (3) full text not available; (4) health science students other than medicals (nurses, dental students, etc.).

### 2.3. Information Sources and Search Strategy

Three electronic databases (PubMed, Web of Science (WoS), Scopus) with no restriction of language, race, age, or publication period were used. Articles published between January 2020 and December 2021 were included. January 2020 was elected due to the fact that clinical trials of vaccines were already underway with the possible participation of medical students, which could be published. Medical Subject Headings (MeSH) and key words were used together using “OR” and “AND”, including COVID-19 OR SARS-CoV-2 OR 2019-nCoV OR coronavirus AND vaccination OR immunisation OR vaccine AND medical students OR medical school OR medical education OR medical undergraduate (Appendix A).

### 2.4. Selection Process

Two independent researchers extracted literature data according to key phrases. After removing duplicates, articles were searched by title and abstract. Nonrelevant articles, reports, editorials, letters, and protocols were excluded. The obtained full-text articles were analysed for eligibility. Articles that did not include COVID-19 vaccinated medical students were excluded. Any differences in the results were discussed between the two researchers. If necessary, a third researcher was consulted to reach an agreement. The resulting data on the last author’s name, year of publication, type of study, country, study period, number of students, number of vaccinated students, and attitudes of students were included in the Excel table (Microsoft Excel 365, Microsoft Corp., Redmond, Washington, DC, USA).

### 2.5. Study Risk of Bias Assessment

Two independent researchers assessed the risk of bias in the included studies. The quality assessment of selected studies was evaluated by six items from the Downs and Black assessment checklist [14,15]. We evaluated the quality of the study according to the total achieved score: a total score of 5–6 indicated high-quality, 3–4 indicated moderate-quality, and 1–2 indicated low-quality studies (Table 1). Any differences in the results were discussed between the two researchers. If necessary, a third researcher was consulted to reach an agreement.

### 2.6. Outcome Measurements

These were the prevalence and various potential influencing factors of COVID-19 vaccinated/nonvaccinated medical students.

### 2.7. Data Analysis

All analyses were performed with the Comprehensive Meta-Analysis Software (Biostat Inc., Englewood, NJ, USA). The heterogeneity of results among studies was quantified using the inconsistency index I^2^. If heterogeneity was evident (I^2^ > 75%), the random-effects model was used; if not, the fixed-effects model was used. Publication bias was assessed using Egger’s test (funnel plot) (Appendix A).

## 3. Results

### 3.1. Study Selection

Through the PRISMA strategy, a total of 969 articles were obtained (Figure 1). There were only 40 articles deemed to have relevance to the systematic review and meta-analysis. Sixteen articles were excluded because the COVID-19 vaccination of medical students was not included. Seven studies were excluded for focusing on another topic (e.g., only nonvaccinated students; the knowledge and skills acquired through an e-learning course; the association between university curricula and the degree of hesitancy for the COVID-19 vaccine). Eleven studies were excluded because medical students were not COVID-19-vaccinated and future acceptance of vaccination was analysed. Based on the inclusion criteria, six studies were included [5,16,17,18,19,20].

### 3.2. Study Characteristics

Six cross-sectional studies from different countries with 4,118 respondents were included in this meta-analysis. All studies were published between February 2020 and August 2021. The prevalence of COVID-19 vaccination varied from 2.0 to 91.9% (Appendix A).

### 3.3. Meta-Analysis of Prevalence of COVID-19 Vaccination

Due to the significantly high heterogeneity (I^2^ = 99.1%; *p* = 0.000), a meta-analysis was performed using a random-effects analysis (Figure 2). The prevalence of COVID-19 vaccination in the six studies was 61.9% (95% CI, 39.7–80.1%).

### 3.4. Subgroup Analysis

#### Gender

For gender, a fixed-effects model was adopted (I^2^ = 73.4%; *p* = 0.005). From 3622 respondents in five studies, 1937 were vaccinated. There was no statistical difference between males (79.4%) and females (74.2%); the OR was 1.038 (95% CI 0.874–1.223). Male gender was a factor affecting vaccination slightly more often than female gender. The authors Sugawara et al., were excluded from the analysis, due to the missing numbers of those vaccinated by gender (Figure 3).

### 3.5. Year of Study

Two studies were excluded due to missing information about the year of study. Thus, in the subgroup analysis of four studies, among 2001 vaccinated students, there were 46.7% students in first year and second year, and 53.3% were students in third year and higher. The vaccination rate in third year and higher was not statistically significantly higher (OR 2.414; 95% CI, 0.754–7.729). A random-effects model was adopted (I^2^ = 97.5%; *p* = 0.000) (Figure 4).

### 3.6. Compulsory Vaccination

Only the three studies in which attitudes to the compulsory vaccination of HCWs were analysed were included in the subgroup analysis. The attitudes towards compulsory vaccination among HCWs can be determined by a prevalence of 71.4% (95% CI, 67.0–75.4%). A random-effects model was adopted (I^2^ = 78.7%; *p* = 0.009) (Figure 5).

## 4. Discussion

The prevalence of COVID-19 vaccination from this meta-analysis of 4118 medical students was 61.9%; the prevalence varied from 2.0 to 91.9%. This contradiction can be explained by a country’s development status and the variable impact of COVID-19 worldwide. Subsequently, in developed countries such as the USA or Japan, the prevalence was 91.9% and 89.1%, respectively [16,17]. A relatively high prevalence was recorded in Slovakia (71.7%) [5]. On the other hand, in countries such as India and Saudi Arabia, we recorded a lower prevalence of vaccination among medical students (64.5% and 66.2%) [18,19]. The lowest prevalence was recorded in Kazakhstan, at only 2% [20].

Generally, concerns about adverse effects of the COVID-19 vaccine and concerns about vaccine efficacy were highlighted by most studies. Moreover, these are the two most frequently cited reasons for vaccine hesitancy. Vaccine hesitancy means a ‘delay in acceptance or refusal of vaccination despite availability of vaccination services. Vaccine hesitancy is complex and context specific, varying across time, place and vaccines. It is influenced by factors such as complacency, convenience, and confidence.’ [21]. Almost 97% of Egyptian medical students were concerned about the adverse effects and more than 93% were concerned about its effectiveness [22]. We can observe similar results in the studies of other authors [18,20].

A large number of published studies have presented the results of attitudes to the intention to be vaccinated, as they were carried out at the time vaccination began. For this reason, many studies have not been included in our meta-analysis. The prevalence of COVID-19 vaccination in these type of studies varied from 35.9 to 94.6% [4,6,22,23,24,25,26,27,28,29,30]. Approximately 92% of Polish medical students and 77% of American medical students were willing to get vaccinated [6,29]. On the other hand, the majority of Ugandan medical students (62.7%) were not willing to be vaccinated against COVID-19 [25]. Among Italian university students, the intention to get the forthcoming COVID-19 vaccine was 86.1% [31].

There are few studies in the available articles that analysed gender differences. In our subgroup analysis of five studies, there was no statistical difference between males and females, although male gender was a factor affecting vaccination more often than female gender. According to Kanyike et al., (2021), males are twice more likely to take up the COVID-19 vaccine than females [25]. Moreover, this finding has been reported by other studies [32]. The acceptance of the forthcoming COVID-19 vaccine among women and men was 66.8% and 76.1%, respectively [24]. Overall, no significant gender difference was observed between HCWs and the general population. However, based on results of Iguacel et al., (2021), females showed a statistically significantly higher percentage of mistrust compared to males [33].

In addition, the scarcity of clinical practice in medical students attending the first and second years may be related to responses of COVID-19 vaccine hesitancy. The curriculum of first- and second-year students is focused primarily on the level of theoretical knowledge. We noticed in our subgroup analysis that COVID-19 vaccination depended on year of medical study. The vaccination rate in third year and higher was higher than in first- and second-year students. The clinical years of medical study are considered a significant factor affecting pro-vaccination attitudes. According to Gao et al., (2021), medical students who had received relevant training were more willing to receive the COVID-19 vaccines [34]. Moreover, training is typical for the clinic years of study. HCWs as well as medical students and other healthcare students are indicated as a susceptible group to COVID-19. Medical students in several countries help medical teams, so vaccination is a crucial step in preventing infections resulting from work in a high-risk working environment [7,35]. Nevertheless, a large number of medical students or HCWs are still hesitant, and the willingness to vaccinate is still relatively low among these population groups. Compulsory vaccination of selected population groups is currently being introduced in several countries. Some countries require compulsory vaccination against COVID-19 among HCWs and some in the at-risk population, i.e., in people over 60 years of age [36,37,38]. Subgroup analysis of our meta-analysis revealed that 71.4% of medical students from three studies agreed to the introduction of compulsory vaccination for HCWs. Based on data from Jain et al., (2021), more than three-fourths of medical students viewed that the COVID-19 vaccine should be made compulsory for HCWs. Meanwhile, those who hesitated to get vaccinated were less convinced of the usefulness of vaccines [18].

### Strengths and Limitations

This meta-analysis protocol was registered, and it was reported in accordance with the PRISMA guidelines. This is the first meta-analysis to evaluate the prevalence of COVID-19 vaccination among medical students worldwide. The study provides a map of the region’s impact on the vaccination of medical students from different parts of the world. The limitation of this study is due to the limited number of available and published articles on the prevalence of COVID-19 vaccination among medical students.

## 5. Conclusions

The prevalence of COVID-19 vaccination among medical students was at a moderate level of 61.9%. Therefore, intervention programs are needed to increase the prevalence of vaccination. Effective tools could be education, practical skills, and the involvement of medical students directly in practice in first years of studies. This will be important in the fight against the COVID-19 and also in another pandemic.

## Figures and Tables

**Figure 1 ijerph-19-04072-f001:**
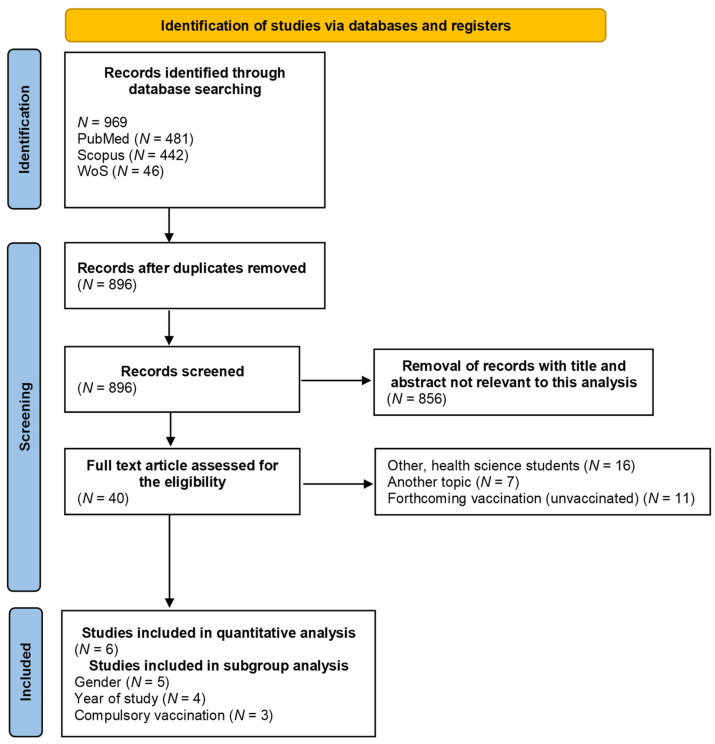
PRISMA flow diagram of screening process.

**Figure 2 ijerph-19-04072-f002:**
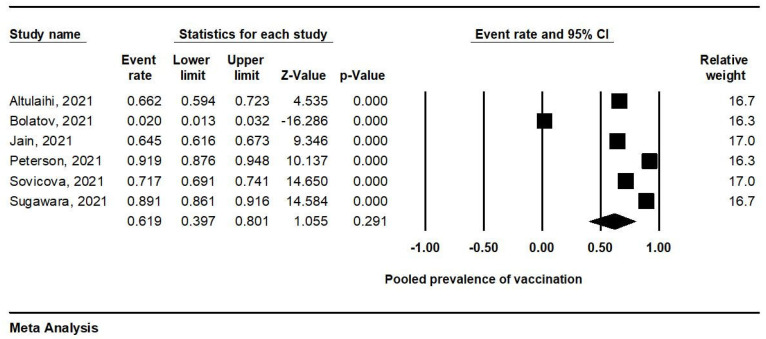
Forest plot of vaccination prevalence.

**Figure 3 ijerph-19-04072-f003:**
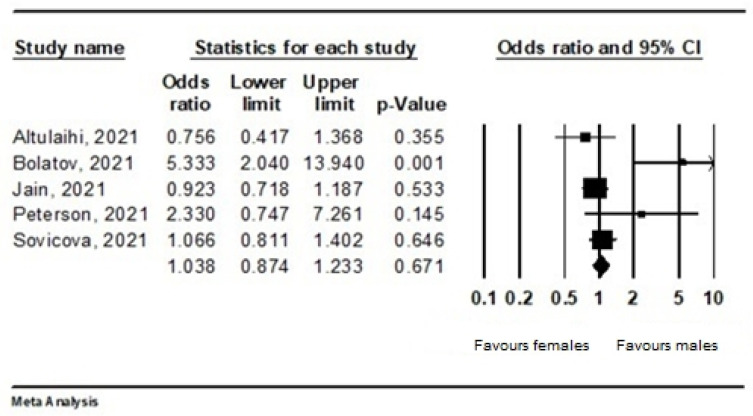
Forest plot representing the relationship between vaccination acceptance and gender.

**Figure 4 ijerph-19-04072-f004:**
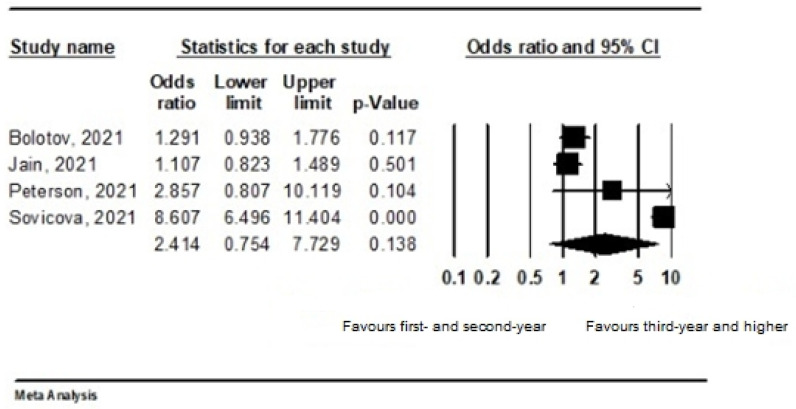
Forest plot representing the relationship between vaccination acceptance and year of study.

**Figure 5 ijerph-19-04072-f005:**
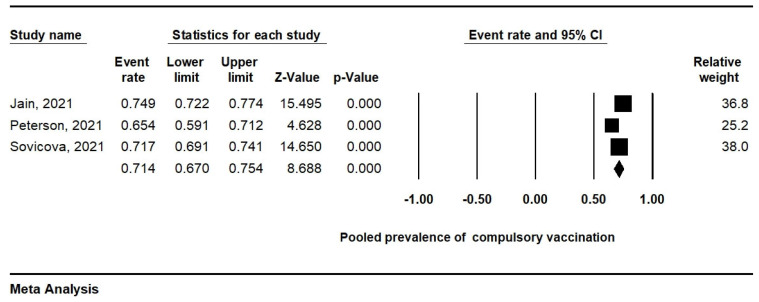
Forest plot representing the relationship between vaccination acceptance and attitudes towards compulsory vaccination of HCWs.

**Table 1 ijerph-19-04072-t001:** Characteristics of studies.

Study, Year	Clearly Stated Aim	Clearly Defined Study Population	Study Sample Representative of the Source Population	Attempt Made Adjusted for Confounding	Attempt Made to Validate Survey Response to Institutional Records Where Possible	Discussion of Study Limitation	Quality Grading
Altulaihi, 2021	Yes	Yes	Yes	Unable to determine	No	Yes	4
Bolatov, 2021	Yes	Yes	Unable to determine	Unable to determine	No	Yes	3
Jain, 2021	Yes	Yes	Yes	Unable to determine	No	Yes	4
Peterson, 2021	Yes	Yes	Unable to determine	Unable to determine	No	Yes	3
Sovicova, 2021	Yes	Yes	Unable to determine	Unable to determine	No	Yes	3
Sugawara, 2021	Yes	Yes	Unable to determine	Unable to determine	No	Yes	3

## Data Availability

All data are fully available without any restriction upon reasonable request.

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
