# Peer review of "Prevalence of COVID-19 Vaccination among Medical Students: A Systematic Review and Meta-Analysis"

_ijerph, 2022, doi:10.3390/ijerph19074072_

Round 1

Reviewer 1 Report

line number 15 and 69:
it is stated that the works from January 2019 to January 2021 were analyzed. Why this period? The vaccine was not available until later, in 2020.  (A typo)? 

This is the first meta-analysis of medical students' attitudes to evaluate the prevalence of COVID-19 vaccination.  Protocol of meta-analysis was registered and approved in Prospero database. Authors performed the meta-analysis according to the PRISMA guidelines. The meta-analysis, despite the low number of available studies (6), reflects the current situation and differences in the prevalence of COVID-19 vaccination among medical students worldwide (Saudi Arabia, Kazakhstan, India, US, Slovakia, and Japan). The results of the meta-analysis point to cultural and political differences in vaccination issues. The study is innovative for further education of medical students in the field of vaccination.  

Author Response

Dear reviewer,

thank you for your comments, which were very inspiring for us. Your comments are very helpful for the improvement of our manuscript. We have edited the text of the manuscript according to your recommendations.

Point 1: line number 15 and 69: it is stated that the works from January 2019 to January 2021 were analyzed. Why this period? The vaccine was not available until later, in 2020.  (A typo)?

Response 1: Thank you very much for reporting an error in the date - the correct date should be January 2020. The reason was the fact that clinical trials of vaccines were already underway with the possible participation of medical students, which could be published.

We have also added this explanation to the methodology (January 2020 was elected due to the fact that clinical trials of vaccines were already underway with the possible participation of medical students, which could be published.).

Reviewer 2 Report

The article is well organized and contains all the elements representative of
a systematic review and meta-analysis.
The aim of the work is clearly defined, however, its limited only to the prevalence of COVID-19 vaccination among medical students.  In a systematic review of such a scope factors determining students’ decisions about vaccination are worth considering.

The results are synthetically presented in a descriptive and graphic manner. I recommend supplying figures 1-5 with legends under the figures, which may significantly increase their readability.

The first conclusion seems relevant to the purpose of the study, but the proposed interventions do not correspond with the obtained results and presented review of the literature on the subject. The choice of the methods would be justified if such variables as, for instance, the level and source of students’ knowledge about Covid – 19 vaccinations were analysed.
Hence, the practical implications are, in my opinion, not very convincing.

Nevertheless, the manuscript  contributes significant cognitive value to medical science

Author Response

Dear reviewer,

thank you for your comments, which were very inspiring for us. Your comments are very helpful for the improvement of our manuscript. We have edited the text of the manuscript according to your recommendations.

Point 1: The aim of the work is clearly defined, however, its limited only to the prevalence of COVID-19 vaccination among medical students.  In a systematic review of such a scope factors determining students’ decisions about vaccination are worth considering.

Response 1: Thank you for your comment. The aim of this meta-analysis was to evaluate only the prevalence of COVID-19 vaccination among medical students worldwide.

Point 2: I recommend supplying figures 1-5 with legends under the figures, which may significantly increase their readability.

Response 2: Thank you for your comment. We used the international standard for describing meta-analysis in figures.

Point 3: The first conclusion seems relevant to the purpose of the study, but the proposed interventions do not correspond with the obtained results and presented review of the literature on the subject. The choice of the methods would be justified if such variables as, for instance, the level and source of students’ knowledge about Covid – 19 vaccinations were analysed. Hence, the practical implications are, in my opinion, not very convincing.

Response 3: Thank you for your comment. We adjusted the conclusion as recommended.

The prevalence of COVID-19 vaccination among medical students was at a moderate level of 61.9%. Therefore, intervention programs are needed to increase the prevalence of vaccination. Effective tools could be education, practical skills, and the involvement of medical students directly in practice in first years of studies. This will be important in the fight against the COVID-19 and also in another pandemic.

Reviewer 3 Report

Part Material and Methods

Eligibility criteria:

  1. You also took into account the number of doses of vaccine, ie. was it a complete or incomplete vaccination?
  2. Why were other students (dental students, nurses) excluded from the study?

Information sources and search strategy:

  1. You declare that you did not have a language limit when searching for articles. It would be appropriate to add the language of the studies included and their number.
  2. On what basis did you choose the search interval January 2019 - December 2021 (the disease was first reported in December 2019 and the first vaccines began to be administered worldwide at the end of 2020)?

Author Response

Dear reviewer,

thank you for your comments, which were very inspiring for us. Your comments are very helpful for the improvement of our manuscript. We have edited the text of the manuscript according to your recommendations.

Point 1: You also took into account the number of doses of vaccine, ie. was it a complete or incomplete vaccination?

Response 1: Thank you for your comment. The inclusion criteria were any vaccination, not the number of doses of vaccine - even in the studies these data were not mentioned.

Point 2: Why were other students (dental students, nurses) excluded from the study?

Response 2: Thank you for your comment. The aim of this meta-analysis was to evaluate the prevalence of COVID-19 vaccination among medical students worldwide to ensure the homogeneity of the target population.

Point 1: You declare that you did not have a language limit when searching for articles. It would be appropriate to add the language of the studies included and their number.

Response 1: According to the current scientific literature, only studies that meet the study criteria are analysed in meta-analysis, regardless of the language. Three electronic databases with no restriction of language were used, however, all articles in our final analysis were in English.

Point 2: On what basis did you choose the search interval January 2019 - December 2021 (the disease was first reported in December 2019 and the first vaccines began to be administered worldwide at the end of 2020)?

Response 2: Thank you very much for reporting an error in the date - the correct date should be January 2020. The reason was the fact that clinical trials of vaccines were already underway with the possible participation of medical students, which could be published.

We have also added this explanation to the methodology (January 2020 was elected due to the fact that clinical trials of vaccines were already underway with the possible participation of medical students, which could be published.).
